# *Living in the Dark:* MQTT-Based Exploitation of IoT Security Vulnerabilities in ZigBee Networks for Smart Lighting Control

**Noon Hussein** [1,2,*] **and Armstrong Nhlabatsi** [3]

1   Department of Electrical and Computer Engineering, University of Waterloo, Waterloo, ON N2L 0K5, Canada
2   Department of Electrical Engineering, Qatar University, Doha P.O. Box 2713, Qatar
3   KINDI Center for Computing Research, Qatar University, Doha P.O. Box 2713, Qatar
*   Correspondence: n5hussei@uwaterloo.ca

**Abstract:** The Internet of Things (IoT) has provided substantial enhancements to the communication of sensors, actuators, and their controllers, particularly in the field of home automation. Home automation is experiencing a huge rise in the proliferation of IoT devices such as smart bulbs, smart switches, and control gateways. However, the main challenge for such control systems is how to maximize security under limited resources such as low-processing power, low memory, low data rate, and low-bandwidth IoT networks. In order to address this challenge the adoption of IoT devices in automation has mandated the adoption of secure communication protocols to ensure that compromised key security objectives, such as confidentiality, integrity, and availability are addressed. In light of this, this work evaluates the feasibility of MQTT-based Denial of Service (DoS) attacks, Man-in-the-Middle (MitM), and masquerade attacks on a ZigBee network, an IoT standard used in wireless mesh networks. Performed through MQTT, the attacks extend to compromise neighboring Constrained Application Protocol (CoAP) nodes, a specialized service layer protocol for resource-constrained Internet devices. By demonstrating the attacks on an IKEA TRÅDFRI lighting system, the impact of exploiting ZigBee keys, the basis of ZigBee security, is shown. The reduction of vulnerabilities to prevent attacks is imperative for application developers in this domain. Two Intrusion Detection Systems (IDSs) are proposed to mitigate against the proposed attacks, followed by recommendations for solution providers to improve IoT firmware security. The main motivation and purpose of this work is to demonstrate that conventional attacks are feasible and practical in commercial home automation IoT devices, regardless of the manufacturer. Thus, the contribution to the state-of-the-art is the design of attacks that demonstrate how known vulnerabilities can be exploited in commercial IoT devices for the purpose of motivating manufacturers to produce IoT systems with improved security.

**Keywords:** ZigBee; CoAP; MQTT; IoT; vulnerabilities; attacks; countermeasures; smart bulb; home automation

## 1. Introduction

The Internet of Things (IoT) describes sensor-embedded physical objects with processing ability, software, and associated technologies to connect and exchange data with other systems over the Internet or communications networks. Since IoT networks allow devices to connect over the Internet, this potentially exposes them to serious threats if not properly secured. Presently, a major challenge is maintaining high performance while ensuring the security of the data exchanged by connected devices and networks in IoT. Owing to Internet-supported connectivity, IoT networks have a large attack surface, which motivates threat actors to remotely exploit vulnerabilities in those networks. Aside from that, IoT devices are usually of limited computing power, which constrains the integration of sophisticated firewalls or anti-malware software. Therefore, the security of information in IoT networks must consider a large number of attack entry points.

ZigBee is an IEEE 802.15.4-based specification for high-level communication protocols that enables the creation of small, low-power, low data rate, low-bandwidth, and close proximity digital radio networks, such as those created for home automation applications [1]. As one of the most popular IoT protocols [2], ZigBee is specialized for building home automation applications. The protocol can be found in IKEA Smart Lighting, Philips Hue Lights, Xiaomi Mijia, and many other smart home solutions. Secured by symmetric encryption keys, ZigBee security is based on the assumption that keys are securely stored and never transmitted unencrypted. In that respect, the extraction of data packets containing such sensitive information by packet sniffing tools exploits the IoT protocol and exposes all network elements to adversary attacks. In view of this, a smart lighting system was selected for experimentation to demonstrate the practicality of proposed attacks on the popular IoT protocol and the products it supports.

The Constrained Application Protocol (CoAP) is a Web transfer protocol based on the User Datagram Protocol (UDP) that is specialized for use with constrained nodes and IoT networks [3–5]. The design of CoAP enables the joining of simple constrained devices to the IoT network, albeit their low power, low bandwidth, and low availability. At its simplest, this packet-based radio protocol is similar to the HyperText Transfer Protocol (HTTP) with GET and PUT commands, and allows communication over various network topologies for low-cost, battery-operated devices at significantly low power levels.

Safeguarding connected devices and networks in ZigBee and CoAP-based IoT systems is a major concern. The attacks described in Section 4 and performed in Section 5 are, partly, made possible by KillerBee; a framework and tools for testing and auditing ZigBee and IEEE 802.15.4 networks. Using KillerBee, network key retrieval when in proximity to the nodes, manipulation and injection, network sniffing, traffic replay, and cryptosystem attacks are only a few possible attacks [6]. Based on the discovery of ZigBee-enabled networks and devices, performing attacks on a ZigBee/CoAP network was possible through the retrieval of sensitive information, in addition to the falsification of the gateway IP address directly connecting the end user and ZigBee End Device (ZED).

In this work, three remote attacks are presented for the purpose of taking control of a ZigBee/CoAP network. The attacks have been tested on the IKEA TRÅDFRI lighting system to evaluate their feasibility and efficiency. The results demonstrated on the system prove the possibility of successful execution of Denial of Service (DoS), Man-in-the-Middle (MitM), and masquerade attacks on a target node, then spreading the attack to compromise all ZigBee and CoAP nodes with Internet connectivity. Considering previous works on ZigBee and CoAP attacks and mitigation techniques, the contribution of this paper is twofold. Vulnerability Exploitation —the design and implementation of three MQTT-based attacks exploiting ZigBee and CoAP vulnerabilities. Namely, through the MQTT protocol, DoS, MitM, and masquerade attacks are performed and evaluated on a popular IoT smart home system. Attack Mitigation—finally, this work is distinct from others in the sense of proposing both the design and implementation of attacks in addition to countermeasures as compared to limiting the scope to either topic. Specifically, MQTT-based detection of message payloads is used to determine the security status of the network in the DoS attack. In addition, the safekeeping of IP and MAC addresses in a database to authenticate users through an MQTT gateway is used to mitigate MitM and masquerade attacks. As such, this work shows that the attacks and mitigations generalize to ZigBee-enabled systems, which is a notable contribution to the state of the art.

We note that this project was not in partnership with IKEA, as there was no agreement between the authors and the company, and they were not notified before the execution of the experiments. After generating the results and proving the vulnerabilities exploitable, however, IKEA Canada was contacted and notified of the vulnerabilities and performed attacks. The response showed little concern for the security of the popular IoT protocol used in their products, as technical specialists merely suggested reporting the issue to customer service.

The rest of this paper is structured as follows: Section 2 provides security analyses of

ZigBee, CoAP, and MQTT protocols. Section 3 presents a comprehensive look into recent related work (2019-2022) exploiting ZigBee and CoAP networks. Section 4 expounds on the attacks, consisting of DoS, MitM, and masquerade attacks, while Section 5 presents and evaluates the feasibility of the proposed attack scenarios. To detect and mitigate the proposed attacks, Section 6 presents the design, implementation, and evaluation of countermeasures along with a few recommendations to secure IoT communication, whereas Section 7 reflects on the performed attacks. Finally, Section 8 summarizes key findings of the paper and emphasizes the persistent challenge of securing IoT devices and networks, followed by future improvements to fully utilize and optimize current and future designs covered in this work.

## 2. Background: Security in ZigBee, CoAP, and MQTT

### 2.1. ZigBee Protocol

#### 2.1.1. ZigBee Overview

ZigBee is an IEEE 802.15.4-based specification for a number of high-level communication protocols used to create personal area networks. Characterized by very low power consumption and low data transfer rates, ZigBee is an adequate protocol for embedded system applications, given its capability of minimizing the frequency of battery consumption and replacement, and provision of a communication coverage of a radius as large as 1000 m at a rate up to 250 kbps [6].

As seen in Figure 1, specifications of ZigBee 3.0 device types include three logical node types: ZigBee Coordinator (ZC), ZigBee Router (ZR), and ZEDs [7]. Exactly one ZC is found in each ZigBee centralized network, as shown in Figure 1a. It is a router with gateway functions that coordinates network actions and controls network bootstrapping. During bootstrapping, the ZC selects the Personal Area Network (PAN) identifier to be used by the network, in addition to the physical radio channel of the network. After that, the ZC acts as a routing device. ZEDs represent sensor nodes and are usually battery-powered. They are responsible for finding and joining the correct network, polling parents to check if messages were sent while it was in sleep mode, and finding new parents if links to old parents are lost [8].

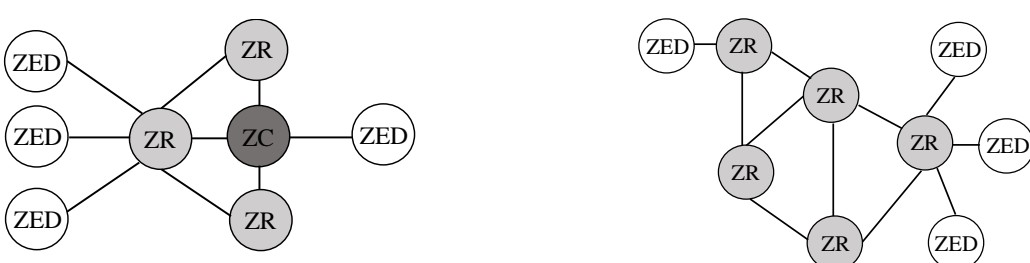

(**a**) Centralized ZigBee network.                                    (**b**) Distributed ZigBee network.

**Figure 1.** ZigBee 3.0 device types in centralized and distributed networks.

Since they are battery-powered, ZEDs are mostly in sleep mode when not in use by applications, but wake up periodically to communicate with other nodes. A ZR is responsible for finding and joining the correct network, perpetuating broadcasts across the network, discovering and maintaining routes, allowing permittable devices to join the network, and storing packets on behalf of sleeping children. A ZR builds a network with other ZRs to exchange packets and can be logically attached to one or more ZEDs. To communicate with ZEDs, ZRs buffer data sent to their ZEDs and send such data at poll requests from ZEDs, while ZEDs send data to ZRs with no request requirements, since ZRs are always awake.

ZigBee 3.0 can form either *distributed* or *centralized* networks. In Figure 1a, every centralized security network is managed by one coordinator. Authenticating and joining new nodes to the network, it is considered the Trust Center in the centralized network. In

Figure 1b, a distributed ZigBee network is shown. There is no coordinator in a distributed network and the security network is formed by a router and end devices. An arbitrary router authenticates and joins new nodes to the network, which makes the authenticating router their parent node. It is to be noted, however, that this simpler model makes systems less secure.

### 2.1.2. ZigBee Security Features

A symmetric key is shared between every two communicating ZigBee devices. An Advanced Encryption Standard (AES) block cipher of a 128-bit key is used to secure the communication [9]. The main security features of the ZigBee protocol include:

1.  *Centralized Trust Center*: in a secured ZigBee network, the ZC node usually authenticates devices attempting to join the network and distributes security keys to authenticated nodes. On the other hand, routers form the distributed network and are responsible for joining other ZRs and ZEDs. Likewise, routers issue security keys to newly joined ZRs and ZEDs.
2.  *Network key security*: when joining the network, the Trust Center shares a randomly generated encryption key with the new device that is common to all network nodes. This network key is used for general protocol maintenance data exchange, and in other applications for user data encryption/decryption. Known to the Trust Center and node, a pre-configured key is used to encrypt the network key after the distribution. Despite this, there remains the huge risk of having a single shared network encryption key in the ZigBee protocol, since obtaining the key will enable threat actors to extend their attacks and compromise all network nodes.
3.  *Link key security*: it is also possible for two nodes to share a unique link key for their communication provided by the Trust Center. Utilizing a link key, two nodes are provided with application level security in addition to that of the network key.
4.  *Certificate-Based Key Establishment (CBKE)*: a CBKE is employed by some ZigBee application profiles to derive unique keys. In this sense, each device must store a certification issued by a trusted authority. For instance, Smart Energy Profile refers to the Elliptic Curve Qu-Vanstone (ECQV) Implicit Certificate Scheme [10]. Obtaining a certificate is a prerequisite for generating a public key as well as other security features and elements.
5.  *Network Layer security*: the Network Layer receives keys and frame counters from upper layers and establishes its own security layer. In addition, route request messages are sometimes broadcasted, and received replies are processed using link keys or an active network key.
6.  *Application Layer security*: securely processes frames before transmission, and establishes and manages cryptographic keys through receiving issued primitives from other layers. An addition to the Zigbee 3.0 security is the ability to create a secure link between two devices in this level through an AES-128 encryption key set between the pair.

In previous ZigBee versions, the usage of a symmetric encryption key to distribute unique network keys to new devices joining the network was a huge vulnerability. A replacement of symmetric keys has been a significant security improvement in ZigBee 3.0. Specifically, the new version replaces the symmetric key with a unique joining key generated from a device-specific installation code. The installation code can be printed on the device or an out-of-band method of passing the code from the device to the ZC, such as NFC or Bluetooth Smart.

### 2.2. CoAP Protocol

### 2.2.1. CoAP Overview

CoAP was introduced to overcome HTTP limitations and provide a standardized alternative to the proprietary protocols [11]. It is a specialized Web transfer protocol used with constrained IoT nodes and networks. Designed to meet specialized requirements,

CoAP offers simplicity, low overhead, and Machine-to-Machine (M2M) and Machine-to-Customer (M2C) communication for less memory and power requirements. In addition, it mandates minimal overhead due to operating over UDP, which allows extended sleep states and faster wake-up times and offers smaller packets for faster communication cycles. Among others, the key features of CoAP are M2M communication; security binding to DTLS; asynchronous message exchange; low overhead and parsing complexity; Uniform Resource Identifier (URI); content-type support; built-in resource discovery; simple proxy and caching support; and UDP binding with optional reliability supporting unicast and multicast requests [12].

### 2.2.2. CoAP Security Features

Integrating Transport Layer Security (TLS), DTLS, HyperText Transfer Protocol Secure (HTTPS), or any third-party security algorithm is necessary to improve CoAP security. Most commonly accomplished by DTLS, security binding in the Transport Layer and AES/CCM offered by the two-layer protocol provides confidentiality, integrity, and authentication [13]. The bottom layer contains the Record protocol, whereas the top layer includes the Alert, Handshake, and Application Data protocols. The Change Cipher Spec protocol messages may also replace either protocol and are used to notify the bottom layer protocol to protect subsequent records [14]. DTLS solves reordering and packet loss problems by adding implementation for packet re-transmission, sequence number assignment within handshake, and replay detection. The protocol also avoids cryptographic overhead issues in lower-layer security protocols. Object Security for Constrained RESTful Environments (OSCORE) is an extension to CoAP and reuses CoAP's serialization of messages to enable end-to-end security at the application layer in the presence of malicious proxies [15].

### *2.3. MQTT Protocol*

### 2.3.1. MQTT Overview

Message Queue Telemetry Transport (MQTT) is a lightweight publish/subscribe message transport protocol that usually runs on TCP/IP or any network protocol that supports ordered, lossless, bi-directional connections [16]. Although the naming of this protocol suggests message queuing, there is no message queuing in the publish-and-subscribe protocol [17]. Given their similar implementation stacks, the primary difference is the use of TCP in the transport layer as opposed to UDP in CoAP. MQTT was widely used in oil pipeline monitoring within Supervisory Control and Data Acquisition (SCADA) industrial control systems, as it offered a bandwidth- and energy-efficient solution to remote monitoring [18].

According to Figure 2, MQTT defines two types of network entities: a message broker and a client. The broker is a server that receives and routes client messages to their specified destinations. On the other hand, a client is any device that runs an MQTT library and connects to a broker over a network. Instead of using the address of the recipient, MQTT employs a subject line called a "topic", in which all subscribed clients will receive a copy of the associated message [19]. In the same manner, multiple brokers can publish topics to a specific subscriber. Every client can obtain, transmit, or both obtain and transmit data by publishing and subscribing. When there are new data to distribute, a publisher sends a control message with that data to its broker. After that, the broker distributes the information to all clients subscribed to that topic. As a result, sensitive information will only be retained by the broker. The main advantages of an MQTT broker include the elimination of vulnerable or insecure client connections; ease of scalability; management and tracking of all client connection states; and reduction of network strain without security compromises.

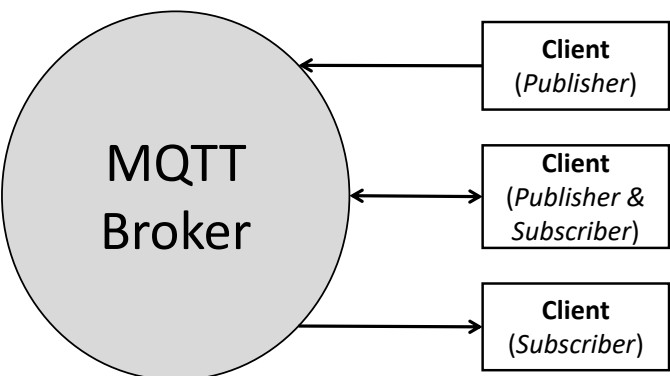

**Figure 2.** MQTT architecture.

2.3.2. MQTT Security Features

MQTT security is enabled at the Network, Transport, and Application Layers, and is a commonly accepted standard mechanism [20]. In the Network Layer, MQTT recommends connecting devices through a gateway, then using a Virtual Private Network (VPN) to connect the gateway to the broker. MQTT uses TLS/SSL in the Transport Layer to encrypt data packets. The standard protocol allows packet encryption, client provision as well as server authentication by client and server certificates. In the Application Layer, MQTT provides both client authentication and data encryption. With the aid of a client ID, username, and password credentials, clients and devices can be authenticated at the broker side.

## 3. Related Work

### 3.1. ZigBee Vulnerabilities and Attacks

Vidgren et al. [21] demonstrated the destruction of ZEDs by sending maliciously crafted signals which constantly awakened them until the battery was fully drained. Since sensors under attack were forced to reply to threat actors, sleep was delayed and the battery discharged quickly. In addition, the work in [22] exploited the vulnerability of ZigBee networks to Low-Rate DoS (LDoS) in indirect transmission, where simulation experiments showed that such attacks could decrease the normal packet arrival rate to 0% through attack timing adjustment.

To detect attacks on the network, especially those utilizing spoofed devices, the authors of [23] analyzed various parameters of the ZigBee (802.15.4) protocol. From the identified parameters, a unique fingerprint of devices was formed, and a white-listed device database was created to keep track of authorized devices, preventing others from joining the network until further authentication is provided. HiveGuard [24], a distributed system, provides archiving, aggregation, inspection, visualization, and alert services to continuously monitor traffic for potential security issues. By testing the open-source software against four energy depletion attacks, it successfully generated an alert for all attacks, which involved selective jamming and spoofed packet injections.

Yang et al. [25] introduced Absolute Slot Numbers (ASNs) and Time Synchronization Tree (TST) attacks, in which time was split into fixed-length slots or ASNs and could target networks of high reliability. Given that an attacker could provide incorrect ASN values to nodes by sending a broadcast message on the network, nodes were incapable of communicating on the network due to such interruption, while an attacker was able to send bogus Directed Acyclic Graph Information Information Object (DIO) packets to desynchronize connections within a network. According to the ZigBee specification, frame counters are sufficient to mitigate replay attacks in secure ZigBee networks. Nonetheless, the work in [26] demonstrated that the network becomes insecure after the coordinator restarts. To mitigate these attacks, the authors presented a power-efficient timestamp-based scheme designed for all ZigBee topologies and different ZED states.

Multiple smart home network security breaches were performed by initially attacking ZigBee devices. For instance, attackers may extrapolate stored or hard-coded network keys in the device by performing firmware dumps [27]. Ronen et al. [28] presented a ZigBee Light Link attack, which stemmed from a worm automatically infecting adjacent bulbs and was deployed as an unauthorized Over-The-Air (OTA) update. Furthermore, WazaBee, a pivotal attack aiming to hijack Bluetooth Low Energy (BLE) devices was implemented in [29], which utilized the compatibility between the two modulation techniques used by ZigBee and BLE.

Through the implementation of a parallel spoofing system, SamBee [30] spoofed ZigBee devices operating in two different frequency channels or jammed such devices operating in five distinct channels simultaneously using a single Wi-Fi frame only. Consequently, maximal corruption was caused in terms of communication links through parallel spoofing and multiple-channel. Another attack proposed in [31] utilized an in-home device and event identifier from encrypted wireless ZigBee traffic. Accessible even from outside the smart home, the identifier inferred a single Application Layer command in the traffic burst of the event, then exploited its periodic reporting pattern and interval. Consequently, attackers could assess the vulnerability of the smart home to unauthorized entry.

On the contrary, two defense strategies were proposed in [32] with the help of a Wi-Fi router. The passive strategy focused on misleading the ZigBee signal sniffing, whereas the active approach distinguished between a common ZigBee signal and an emulated signal in real time. Additionally, the work in [33] proposed a ZigBee receiver to decode signals in the presence of a constant jamming attack by leveraging Multiple-Input Multiple-Output (MIMO) technology. The machine learning-based enabler mitigates unknown interference using an optimized neural network in the face of 20 dB stronger jamming attacks.

In [34], it was demonstrated that most attacks against Zigbee-based IoT devices were dictionary attacks, used by intruders to access the IoT system by trying all character combinations in the dictionary to break the security. On that account, the work in [35] reviews multiple Intrusion Detection Systems (IDSs) for smart grids, including a protection algorithm based on machine learning. The Home Area Network Intrusion Detection and Prevention System (HANIDPS) algorithm compared analyzed and normal traffic to identify threats. Likewise, the exploited Remote ATtention (AT) Commands in [36] allowed a malicious user to reconfigure or disconnect targeted sensors from the network, which motivated the development of an IDS able to detect and protect the devices from the proposed threat.

From the ZigBee 3.0 assessment in [37], three core security issues were investigated: (a) symmetric key security, relating to how an attacker could obtain symmetric keys; (b) compromised symmetric keys, concerning the breach against network confidentiality if one or more symmetric keys are exposed; (c) insufficient DoS protection mechanisms, enabling ZigBee to be susceptible to specific DoS attacks.

### 3.2. CoAP Vulnerabilities and Attacks

The security of this application layer protocol was investigated by Rahman and Shah [38] over Datagram Transport Layer Security (DTLS). Specifically, the challenge of requiring another encrypted protocol, mainly DTLS, to provide security is a huge limitation that also introduces other issues, including large message and handshake compression as well as DTLS compatibility issues with CoAP proxy modes. Nonetheless, an Amplified Reflection Distributed DoS (AR-DDoS) aims to magnify the amount of injected malicious traffic while obscuring the sources of the attack traffic. In [39], this attack was performed on CoAP over IPv4 and IPv6, where the results characterized reflector saturation at low probe injection rates.

Since CoAP uses UDP as a transfer protocol, UDP attacks are inherent in CoAP. In [13], a client–server architecture of CoAP-based end devices and a proxy system across the client side was implemented. Following that, an active interception between client and server ends was launched based on network sniffing, aiming to gain intelligence on the shared

information between both ends. To launch an off-path attack, the authors of [40] analyzed and exploited IP spoofing vulnerabilities and the CoAP remote server access, as a fake request message was injected to modify the IPv6 over Low-Power Wireless Personal Area Network (6LoWPAN) credentials of a smart door keypad lock system. By deploying a machine learning-based approach, it was demonstrated that the proposed attack could be prevented.

Finally, several papers provided mitigation techniques for CoAP networks. One paper in [41] provided mitigation techniques for UDP flood attacks in CoAP using the Cooja simulator, while [42] supported the Authentication and Access Control scheme for CoAP (AAC-CoAP), which significantly improved the CPU calculation time as well as execution time. Nevertheless, the design of an intermediary in [3] incorporated authentication with the Extensible Authentication Protocol (EAP) and CoAP, particularly analyzing the usage of a CoAP proxy as well as introducing CoAP relays and stateless proxies.

### 3.3. Comparison to This Work

This paper utilizes the identified ZigBee and CoAP security vulnerabilities to exploit an IoT network by initiating a passive attack to obtain private and sensitive information. The passive attack was executed by exploiting and compromising the ZigBee installation code. The information obtained is then used to perform DoS, masquerade, and MitM attacks on the network using MQTT. By overloading the network with color and/or state (on/off) switching commands, the ZigBee frequency agility feature is compromised. Authorized end users will no longer be able to control the physical system. Through the evaluation of attack scenarios, major threats affecting the two protocols are recognized. As such, this paper additionally presents the design and implementation of MQTT-based countermeasures to protect the IoT networks from similar attack scenarios, including the monitoring of message payloads to detect intrusions, and the safekeeping of plain-text and hashed IP and MAC addresses in a database to authenticate users by an MQTT gateway and block unauthorized messages from reaching ZEDs. Consequently, this work is expansive in the sense of proposing and implementing specific attacks alongside their mitigation techniques.

## 4. Attacks on Smart Lighting

ZigBee makes three main security assumptions: (1) open trust model between layers; (2) safekeeping of security symmetric keys; (3) implementation/employment of security protection [43]. The assumption of an *open trust* model entails that the protocol stack layers trust each other. Hence, cryptographic protection only exists between devices, but not between different layers in a device. This allows for the reusing of cryptography among layers of the same device, as the communication between its different layers is unencrypted for all ZigBee implementations, which generalizes the attacks to all ZigBee-enabled systems, regardless of the manufacturer. The safekeeping of symmetric keys assumption entails that secret keys are not available unsecured outside the device except during the pre-configuration of a new device, where a single key can be possibly sent unsecured, creating a vulnerability for attackers who have access to the device during that time. To demonstrate, if the Network Key changes, the unencrypted Transport Key can thus be sniffed through packet sniffing tools, such as Wireshark. Note that the low-cost nature of the protocol cannot enable it to protect against hardware attacks as will be seen in the proposed attacks that we explored in this work. The employment of security mechanism protection entails that when joining a network, ZRs and ZEDs must adapt to the network-employed security scheme by supporting both centralized and distributed security.

To demonstrate the possibility and feasibility of compromising ZigBee and CoAP, an attack on the IKEA TRÅDFRI lighting system; an economical smart lighting solution integrating different features such as voice, color, and brightness controls was executed [44]. Specifically, the IKEA TRÅDFRI Gateway Kit was used, which contains a gateway, two smart bulbs, and a remote control [45]. As the ZC, the TRÅDFRI Gateway establishes a connection with the LED bulb through ZigBee, for it is the ZED to be controlled by the user.

Initially, the smart bulb is controllable in two non-interfering ways: using the TRÅDFRI Remote Control or the IKEA Home Smart application. To connect to the gateway, the former and latter use ZigBee and CoAP, respectively. Attacks described in this section will include DoS, MitM, and masquerade attacks targeting the IKEA TRÅDFRI smart lighting.

Intrusions will originate from an unauthorized acquisition of the ZigBee installation code, which can be found on the back of the gateway in the form of a 16-digit alphanumerical code as well as a QR code. Next, a C implementation of CoAP alongside a ZigBee2MQTT library will be installed and configured on a Raspberry Pi 3 Model B+. Utilized as an MQTT broker, the broker will make use of the aforementioned installations to overtake the operation of the mobile application, causing it to be ultimately unresponsive in all attacks scenarios.

### 4.1. Hardware Description and Functions

The experimental setup is shown in Figure 3, where the hardware used is described below.

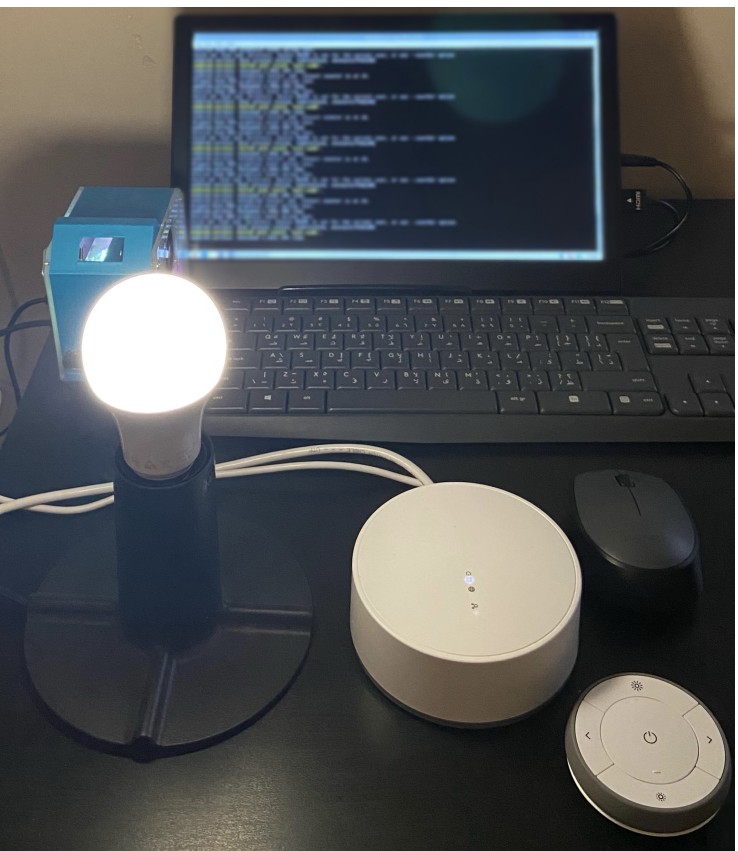

**Figure 3.** Photo of experimental setup showing Rasberry Pi, Smart Bulb, TRÅDFRI Gateway, and Remote Control.

#### 4.1.1. Raspberry Pi 3 B+

A Raspberry Pi 3 B+ is the source of the attacks. It is a Broadcom BCM2837B0, Cortex-A53 (ARMv8) 64-bit system on chip with a 1.4 GHz 64-bit quad-core ARM Cortex-A53 CPU and 1 GB of Low-Power Double Data Rate Synchronous Dynamic RAM (LPDDR SDRAM). The CC2531 packet sniffer will be inserted into the USB port of this device to sniff ZigBee traffic in the network. Running on Linux, a packet sniffing tool will be used alongside the CC2531 USB to decrypt network traffic and obtain the required information. After the attacker gains access to the network, the attacks described in the rest of this section will be initiated.

### 4.1.2. CC2531 USB Dongle

A fully operational USB providing an interface to IEEE802.15.4/ZigBee applications, with a wireless transmission speed rate of 250 Kbaud was used. As it comes preflashed with a CC2531ZNP-Prod firmware, it can be plugged directly and used as a ZigBee packet sniffer. Since the Raspberry Pi uses MQTT, controlling the ZigBee network requires the establishment of a connection between the two networks. As such, ZigBee2MQTT—a nodejs Gateway application—will be used to connect the ZigBee network to MQTT. Through the CC2531 USB dongle integration, ZigBee2MQTT will function as a ZigBee coordinator, creating a ZigBee network and allowing the attacker to reach the ZED through the Raspberry Pi.

### 4.1.3. TRÅDFRI Gateway

The IKEA TRÅDFRI Gateway allows users to connect smart IKEA products such as blinds, light bulbs, and speakers and control them using the IKEA Home Smart application on a phone or tablet. It has a standby power consumption of 0.63 W in the open air and a range of 10 m in the open air. On the device, three light indicators are found:

1. *Power Light*: indicates the power state of the device.
2. *Network Light*: indicates the presence of a TRÅDFRI network. When pulsing, it indicates that no connected TRÅDFRI devices are found.
3. *Internet Light*: there is no Internet connection. When pulsing, it indicates that the user is not properly connected to the Internet.

Through the demonstrated attacks, the only way for the authorized user to reconnect will be by resetting this product to factory settings. This can be done by removing the gateway lid and then pushing a pin into the pinhole on top of the gateway for at least 5 s.

### 4.1.4. TRÅDFRI LED Bulb E27 1000 Lumen

An 11 W, 1000 lumen TRÅDFRI LED bulb was used in the experiment, which has an LED lifetime of about 25,000 h and a standby power consumption of 0.5 W. By connecting to the remote control and/or the gateway, users can switch the TRÅDFRI bulb on or off, adjust its brightness up to 1000 lumens, and change its color between 2200 Kelvin (warm), 2700 Kelvin (normal), and 4000 Kelvin (cool). Alternatively, integrations other than the TRÅDFRI Gateway can be used to connect this smart bulb, such as ZigBee2MQTT, ioBroker, ZiGate, DeCONZ, Zigbee Home Automation, and Tasmota.

Given the above integrations, ZigBee2MQTT will be used in this experiment to establish a connection to the network and maliciously control the smart bulb, therefore gaining control over the authorized gateway. Although power consumption was not recorded throughout the experiment, it is assumed that the continuous malicious commands will increase the consumed power. Since IoT devices save power by being in sleep mode, constantly waking up these devices with commands will result in higher power consumption levels.

### 4.1.5. TRÅDFRI Remote Control

The remote control enables the user to directly control the bulb, independent of a gateway. On the remote control, the following buttons are present:

1. *Power button*: on/off bulb switch. Synchronizes the bulb by pressing and holding for at least 3 s, which returns it to its default setting: 100% brightness at 2700 K.
2. *Dim Up/Down*: dims the brightness of the bulb.
3. *Left/Right arrows*: change bulb colors.

Besides the IKEA Home Smart application, the remote control is another way to control the smart bulb. However, this device will not be directly targeted by the MitM attack, which means the authorized user will still be able to use this device during the attack. In other words, the DoS attack blocks the ZED from receiving commands, whereas the MitM attack

takes commands from the remote control as well as the application as input and alters them.

### 4.2. MitM: Passive Eavesdropping

Since the TRÅDFRI gateway communicates with the IKEA Home Smart application through CoAP, the attack will overtake control over that connection by integrating libcoap; a C implementation of CoAP with Application Programming Interfaces (APIs). In addition, the IP address of the ZC must be obtained to locate the IoT hub. To obtain the IP address of the gateway, an Advanced IP Scanner is to be used to capture the communication in the network and find the IP and MAC addresses of the gateway under "Murata Manufacturing" [46]. Alternatively, the network traffic can be analyzed and the IP and MAC addresses can be found under the same name.

To initiate the attack, the ZigBee installation code found on the back of the gateway must be acquired. Installation code extraction is achievable by eavesdropping on the communication between the device controlling the IKEA Home Smart application and the gateway. Decoding the IEEE 802.15.4-based protocol will be done using a CC2531 USB dongle alongside KillerBee, an IEEE 802.15.4 network packet sniffer, and Wireshark. On channel 11 (0x0B), Wireshark starts capturing ZigBee traffic from the packet sniffer. Because logged ZigBee messages are encrypted, the Trust Center Link Key and the Transport Key must be added. By setting the security level to AES-128 Encryption, 32-bit Integrity Protection on Wireshark, the Trust Center Link Key will be added, which is identical and distributed to all devices on the network.

Next, there are two methods that can be used to obtain the Transport Key: (1) if the Network Key in configuration.yaml has not been changed, the Transport Key will be set to a default value; (2) if the Network Key has been changed, the attacker can retrieve the Transport Key from Wireshark when a ZigBee device joins the network. From the ZigBee Network Layer Data sub-tree, the Key will be unencrypted and exposed to the attacker.

After adding both keys, Wireshark will be able to decrypt ZigBee messages. The installation code is used to encrypt the initial Transport Key from the centralized Trust Center device to the joining device. In ZigBee 3.0, all ZigBee devices capable of joining networks must support installation code usage during joining. On the back of the TRÅDFRI Gateway, the 16-digit alphanumerical installation code ensures parent device authorization to verify that joining devices receive information securely. Although the code can only be captured when a new device joins, the ZigBee 3.0 network does not permit joining forever, as it automatically closes joining after 254 s [7], which expedites the process of installation code extraction by eavesdropping on ZigBee 3.0 communication.

### 4.3. DoS Attack Description

Denial of Service attacks are common in IoT devices [47–49]. The CoAP security analysis presented in Section 2.2 was used to identify the following potential vulnerabilities which were later exploited to launch a DoS attack: *message parsing*; *proxying and caching*; *key generation*; and *cross-protocol exchanges*. *Message parsing*—the improper handling of client/server parsers by the processing logic may affect the availability of CoAP nodes as a result of overloading or the ability to remotely execute code on the node under attack. *Proxying and caching*—the improper implementation of access control mechanisms of proxies and caches may cause confidentiality and integrity breaches by compromising the content of CoAP messages. *Bootstrapping*—the improper implementation of a new CoAP node setup may permit access to unauthorized nodes in the network. *Key generation*—the robustness deficiency of cryptographic key generation may compromise CoAP nodes. *IP spoofing*—the forgery of CoAP nodes' IP addresses allows adversaries to perform a variety of attacks, such as generating spoofed response messages and acknowledgments. *Cross-protocol exchanges*—the response of a CoAP node to a message sent from a spoofed IP address and a fake port number will reach the target node and force it to interpret the received message according to the target protocol.

Following the acquisition of the ZigBee installation code, DoS, MitM, and masquerade attacks will be initiated by using a Raspberry Pi MQTT broker. As seen in Figure 4, the Raspberry Pi broker is to emulate an end user in order to take control of the IoT network. In particular, the IP address and installation code are to be configured in ZigBee2MQTT to gain unauthorized access to the ZigBee network. Next, a CoAP library is to be formed with PSK security enabled. Integrating DTLS, symmetric cryptographic CoAP keys will secure the communication between the broker and the gateway during the attack. To establish communication between the broker and gateway, when a PSK is created, it is then shared in advance with the broker and must be passed to the gateway while connecting to add security. The DoS attack aims to target the IoT system by automatically injecting the ZED with signals to constantly trigger the IKEA smart bulb every 250 milliseconds. As they will target the smart bulb, the payload of TLS/SSL-injected messages will consist of "false" booleans which will flood the ZED and deny any user alterations of the state of the light. As a result, the frequency agility security feature of ZigBee will be compromised, as it allows the Network Manager to change the channel in case the current channel is jammed or has a lot of interference. It should be emphasized that awakening the smart bulb at a high frequency and forcing it to reply to the threat actor delays sleep and negatively impacts the lifetime of the device due to the low polling rate of the IKEA smart bulb compared to the rate at which signals will be injected.

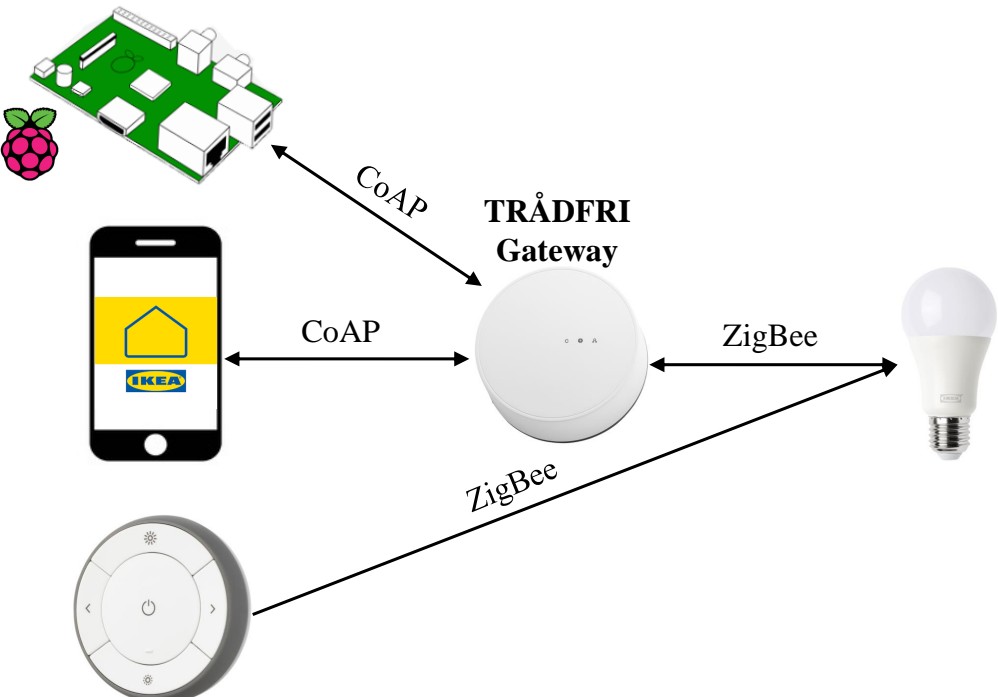

**Figure 4.** DoS attack diagram.

*4.4. MitM and Masquerade Attack Descriptions*

The attack demonstrated in Figure 5 combines an MitM attack—an active attack based on the interception of exchanged data followed by modification or deletion in order to alter the comprehension of the message and prevent a message of integrity from arriving to the receiver—and a masquerade attack—another active attack which uses a fake identity to gain unauthorized access to information through legitimate access identification. Based on collected data from the gateway, the Raspberry Pi will modify the state and color of the lighting. Simultaneously, the remote control will be tricked into believing that changes made to the system are originating from the IKEA Home Smart application. Similar to the previous DoS attack, the MitM and masquerade attacks will ultimately cause congestion of network traffic for the IKEA application, which will lead to the attacker overtaking the

application and denying users from accessing it. Even though the remote will still be of use in that case, MitM and masquerade attacks will continue to alter the input such that the output dissatisfies the user.

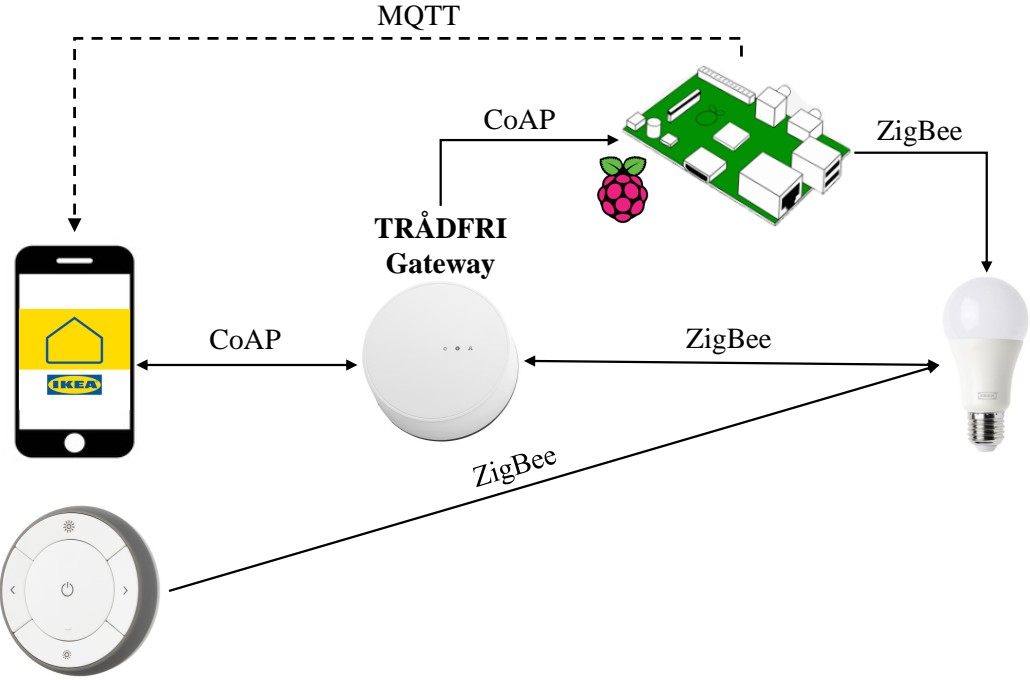

**Figure 5.** MitM and masquerade attack diagram.

In Figure 5, messages received by the gateway from the IKEA Home Smart application are forwarded to the smart bulb. In this case, the gateway is to be intercepted such that the content of each message successfully reaching the smart bulb will be viewed by the attacker, modified, then forwarded to the smart bulb. Specifically, each on/off message showing the status of the bulb will be intercepted at the gateway and triggered such that the output is the inverted user input. At the same time, the chosen bulb color by the user will be altered as follows: (a) warm will be changed to cool; (b) normal will be changed to warm; (c) cool will be changed to warm. Signal interception for payload retrieval was repeated at a regular interval of 1 s, while modification and publish commands will be of no delays. It is worth emphasizing that the switching time of the state and color of the light is limited by the hardware specifications of the manufacturer.

## 5. Results and Evaluation of Attacks

As previously emphasized, unauthorized access on the IoT system will be possible through the acquisition of the ZigBee installation code. Utilizing the CC2531 dongle to sniff ZigBee traffic, Trust Center Key, Transport Key, and Wireshark, the installation code was decrypted, as seen in Figure 6. After obtaining the code, the system was exposed to DoS, MitM, and masquerade attacks implemented in the remainder of this section.

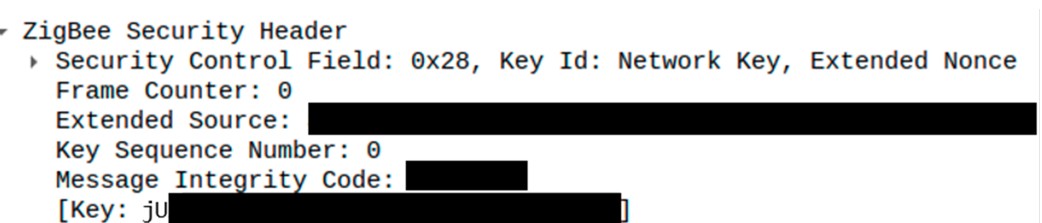

**Figure 6.** Decrypted ZigBee installation code on Wireshark.

The next step after obtaining the code was to join the network through the Raspberry Pi using the installation code. This way, the attacker was masked as a legitimate entity and performed DoS, MitM, and masquerade attacks, as presented in the remainder of this section.

### 5.1. DoS Attack

Taking into account the CoAP vulnerabilities presented in Section 2 VI-B, Figure 7 presents an excerpt of "off" status commands sent to the light bulb through the TRÅDFRI Gateway, following the attack diagram in Figure 4. With each message, the date, time, and payload confirm that the multiple "off" messages were sent per second, according to the selected frequency.

```
21 Oct 17:15:42 - [info] [debug:d269b2340f819ff9]
{
  payload: '{"tstamp":1634825742020,"uptime":15608,"load":[0.44,0.67,0.84],"h
ostname":"raspberrypi","mem":{"used":239712,"free":272736,"swapused":82988,"s
wapfree":19408},"nw":{"wlan0":{"rx":75564755,"tx":61517663}}}',
  topic: '',
  _msgid: 'ed44bfb067fde3a4'
}
21 Oct 17:15:42 - [info] [debug:d269b2340f819ff9]
{
  payload: '{"tstamp":1634825742270,"uptime":15608.25,"load":[0.44,0.67,0.84]
,"hostname":"raspberrypi","mem":{"used":239860,"free":272588,"swapused":82988
,"swapfree":19408},"nw":{"wlan0":{"rx":75565207,"tx":61518496}}}',
  topic: '',
  _msgid: '12203387f1bae213'
}
21 Oct 17:15:42 - [info] [debug:d269b2340f819ff9]
{
  payload: '{"tstamp":1634825742520,"uptime":15608.5,"load":[0.44,0.67,0.84],
"hostname":"raspberrypi","mem":{"used":240108,"free":272336,"swapused":82988,
"swapfree":19408},"nw":{"wlan0":{"rx":75565659,"tx":61519329}}}',
  topic: '',
  _msgid: '609f94e31540513b'
}
```

**Figure 7.** MQTT-based DoS attack on smart bulb.

DoS is a prominent attack that still hinders the functioning of many protocols, including ZigBee and CoAP, thus directly compromising communication efficiency. In contrast with UDP, TCP-based HTTP provides proper end-to-end congestion control, whereas CoAP congestion control is basic considering congestion control in the case of CON messages, which use a fixed Re-transmission Time-Out (RTO) [3]. At first, a random number is assigned to the RTO between a constant ACK_TIME-OUT and a constant ACK_TIME-OUT multiplied by a constant ACK_RANDOM_FACTOR [3]. By means of an exponential back-off mechanism, messages which have not received an ACK within the fixed RTO duration are subject to re-transmission, doubling the value of the RTO. For this reason, the MAX_RETRANSMIT constant defined by CoAP specifies the maximum message re-transmission number before the sender is blocked and the transmission is classified as a failure.

Since CoAP is based on UDP, the reduced QoS forced a CoAP congestion control mechanism on the gateway, terminating the response to incoming messages. However, the physical machine of the attacker overpowered and bypassed this mechanism, which caused the IKEA Home Smart application to try reconnecting, to the gateway as seen in Figure 8a. In that case, the attacker was in full control of the system, and so the application eventually displayed the message in Figure 8b, informing the user of the failed attempt to connect to the gateway and suggesting irrelevant solutions to the issue, which implies that the end-user application had not been designed to detect or defeat DoS attacks.

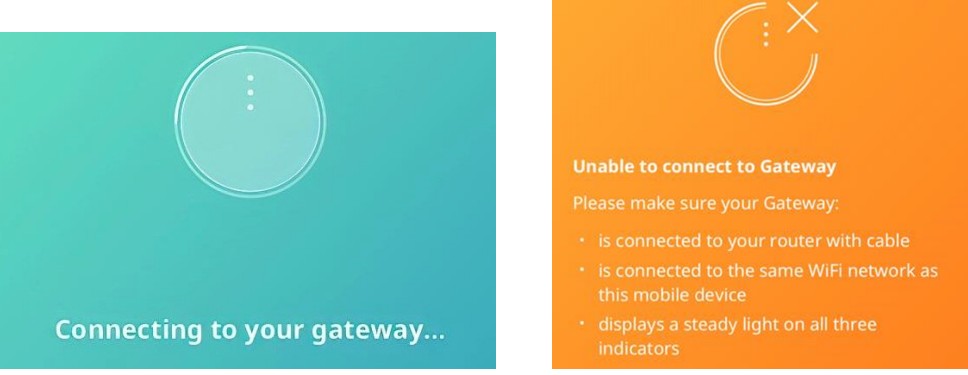

(**a**) Preliminary result                                            (**b**) Final Result

**Figure 8.** Preliminary and final results of network congestion on the IKEA Home Smart application.

Meanwhile, the status of the smart bulb was continuously off, as set to be sent by the attacker every 250 milliseconds, which also disabled the user from trying to turn the lamp on by using the remote due to the high rate of sent messages. It is worth noting that, in the same manner, the attack could be modified to keep the smart bulb on and deny the user from turning it off using the remote or their device running the IKEA application. Nevertheless, Figure 8 above suggests that although CoAP defines a basic congestion control mechanism to independently handle congestion, its behavior was incapable of adapting to network conditions under attacks, making it significantly too aggressive for home automation IoT applications in case of an attack.

*5.2. MitM and Masquerade Attacks*

Based on the attack diagram in Figure 5, the second attack synchronously combined an MitM attack on the application as well as a masquerade attack on the remote control. In particular, while the application was initially operating normally to control the status and color of the light, the broker was configured to capture status and color messages relayed between the gateway and the smart bulb. On that account, the parallel device connection of the attacker was set to operate with a delay of 2 s. That is, every 2 s, every message sent from the application or the remote targeting the light status or color was toggled or changed, respectively, according to the listing in Section 4.4. Since the attacker captured the IP address of the gateway as well as the installation code, it is evident that tricking the user into believing the system was operating normally was initially possible with an MitM attack when no modification actions were taken. In other words, the protocols could not identify the source or type of attack through the exploited vulnerabilities but rather proceeded to block the connection with the legitimate node as if it was malicious. To re-establish an authorized connection to the system, one can only hard reset the gateway, which also requires reconnecting the remote. Even then, the attacker can regain control, as the installation key does not change.

Ultimately, the detection of message re-transmission was possible with the ZigBee MAC, which prevents attacks on encrypted packets where the attacker intercepts, alters, and re-transmits an encrypted message such that it is accepted as legitimate by the receiver. However, the MAC was unable to block maliciously modified messages from the attacker. Instead, the same action in response to DoS was taken, as the application attempted to reconnect to the gateway yet failed to gain control of the firmware over the attacker. Concurrently, since the remote control is merely an input device controlling the bulb without needing a gateway after the setup stage, the device could not receive information such as the status of the gateway or light. As such, the second attack is considered a masquerade attack on the remote control, as it is unaware of malicious data fed into the system, and it falsely assumes that other messages sent to the smart bulb originate from an authorized application user.

The described attacks could be further aggravated by leaving original links and incrementing the version number continuously. Consequently, the gateway would be permanently disabled through reboots and firmware installations. By redirecting Domain Name Server (DNS) requests to the server of the attacker, `version_info.json` could be manipulated to perform firmware downgrade or DoS attacks. All in all, the attacks discussed expose vulnerabilities in ZigBee and CoAP security. From obtaining keys to responding to abnormal behavior, the protocols have proven the need for an update integrating an IDS instead of offering such systems as add-ons to software. In the demonstrated attack, the only way for the user to regain control is to reset the entire system as described in user manual [50]. Even then, the installation code found on the back of the gateway and the IP address of the device would still enable the attacker to resume the attack on the IoT system. Thus, preventing the executed attacks mandates security patching ZigBee and CoAP networks.

## 6. Countermeasure Design and Implementation

Security and privacy are key issues in IoT-based environments, which necessitate IoT IDSs to mitigate attacks. Mainly running in the Network Layer of IoT systems, IDSs are capable of analyzing data packets, generating responses in real time, and adapting to different IoT technologies. Nonetheless, computing and storage limitations of devices and protocols optimally require special IDSs to ensure efficiency, reliability, and robustness, as conventional IDSs may not always be suitable. As discussed, an IoT-based IDS must operate under restrictive conditions of reduced processing capability, fast response, and high-volume data processing. Considering that IoT security is a fundamental and persistent issue, a contemporary understanding of security vulnerabilities facilitates the implementation of corresponding mitigation approaches. On the other hand, the end user must be informed of possible attacks threatening the system in order to properly respond to and/or report them. In the remainder of this section, the previously executed attacks will be detected, mitigated, and the user will be informed of the security breach in the network.

### 6.1. Mitigation of DoS Attack

The detection and mitigation of all demonstrated attacks was possible by replacing the gateway with an MQTT device, as it used embedded CoAP libraries and ZigBee2MQTT to identify the injection of malicious data into the network and prevent it from reaching the ZED. As a more effective approach to detect and prevent the proposed DoS attack, Figure 9 presents a diagram of the implemented IDS, which detects consecutive identical messages, blocks them from altering the state of the bulb, and informs the end user of the detected security breach.

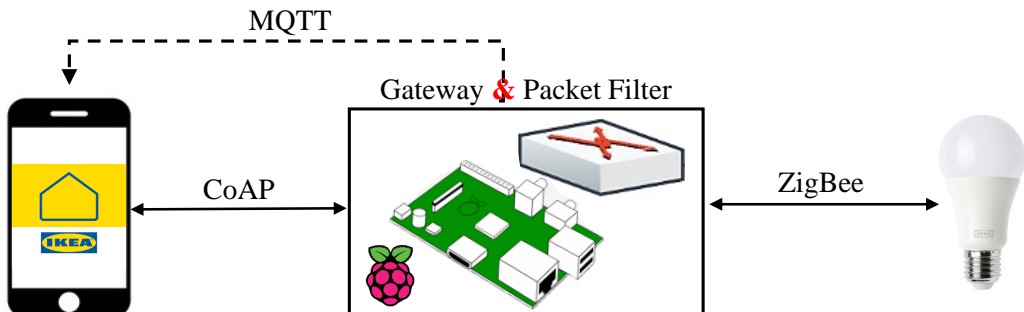

**Figure 9.** DoS attack countermeasure design.

As opposed to disconnecting without providing relevant reasons in the original design, the MQTT device was placed between the end user and ZED to monitor and filter the communication between the two devices, ensuring message confidentiality, integrity, and availability. In particular, the Raspberry Pi analyzed incoming packets from IKEA Home Smart and filtered traffic such that no continuous "off" or "on" messages were sent at

minimum delays to flood the network. That is, since the IKEA Home Smart application and the remote control do not have dedicated "on" or "off" buttons, the single dedicated power button ensures proper switching such that the power sequence behavior does not consist of two consecutive "on" or "off" commands. As a result, the detection of two identical and consecutive state-control messages indicates the existence of a security breach in the ZigBee/CoAP network. To respond to such an attack, it is proposed that the user provides their phone number or e-mail address when setting up the system, such that a message similar to the one displayed in Figure 10 is sent to the user to inform them of a security breach.

## [ALERT: Non-QU Sender] ALERT: IKEA Home Smart Breach

This e-mail is being sent to you because of a security breach that was detected by our system. Our server detected continuous malicious "off" messages received by your smart bulb from an unauthorized source.
Please follow the instructions on the IKEA Home Smart application to ensure your smart home is protected against attacks.
If you do not have the latest version of the application, please upgrade it to reduce chances of malfunctioning.

Thank you for your understanding.

**Figure 10.** User e-mail alert for a DoS Attack.

Alternatively, the device monitoring traffic between the end user and smart bulb can be eliminated by embedding the same algorithm within the IKEA gateway. Unusual traffic detected by the IDS can then be directly accessible to the solutions provider in order to take appropriate action. Despite the fact that an application-embedded IDS would simplify reaching the user, notifying them of a breach, and suggesting adequate solutions, it is up to the solutions provider to update the routing design and add filtering features to the firmware.

### 6.2. Mitigation of MitM and Masquerade Attacks

The simple design of the MQTT protocol in which the message header strictly contains the necessities for delivering a message to a specific topic does not provide room for additional information. Hence, detecting and preventing MitM attacks using the publish/subscribe protocol is not possible, as a key feature of MQTT is disassociating the publisher from the subscriber as best as possible. In essence, the publisher should not be concerned about the number of subscribers in the same way that subscribers should not care about the source of information as long as the received topic matches its subscription.

During the proposed MitM and masquerade attacks, the ZigBee MAC detected the interception, alteration, and retransmission of messages between the ZC and ZED. Since different devices can join the network by installing IKEA Home Smart, the MAC address alone could not be used to identify intrusions. Additionally, the spoofed IP by the attacker does not permit the identification of unauthorized devices using standard IP filtering mechanisms. Instead, combining IP and MAC address verification for each end user joining the network could be realized using MQTT in a two-phase approach based on a packet-filtering firewall that filters plain-text and hashed MAC and IP addresses. The first phase, shown in Figure 11, is authorized device registration, where the addresses (IP and

MAC) of devices whose messages would be allowed are registered in a database of known devices, *K*.

Figure 12 shows the second phase. For each device joining the network, the Address Resolution Protocol (ARP) maps the dynamic IP address to the corresponding MAC address in the *Scan Device* step. The *plaintext* of the combined IP and MAC addresses are represented by $a_p$. Next, the IP and MAC addresses ($a_p$) are hashed to form the corresponding ciphertext, $a_c$. Finally, the packet-filtering firewall allows input messages of authenticated sources to pass to ZEDs, and block/deny those of sources that did not match any elements in *K*. When testing this countermeasure with a malicious MQTT-based device of a spoofed IP, it was observed that input messages sent by the attacker could not reach the smart bulb, which confirms the successful implementation of the database packet-filtering countermeasure.

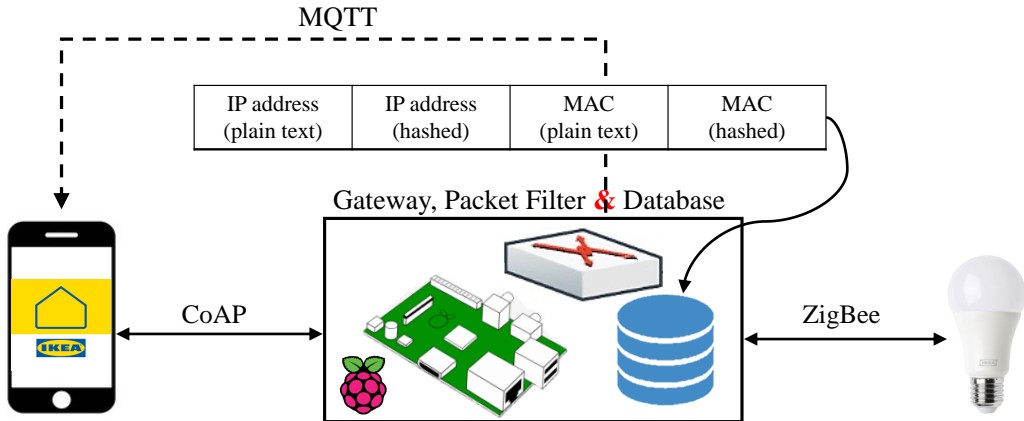

**Figure 11.** Authorized device registration through IP and MAC addresses. Illustrated is an MQTT broker containing a packet filter and a database of plain-text and hashed IP and MAC addresses to authenticate joining ZEDs.

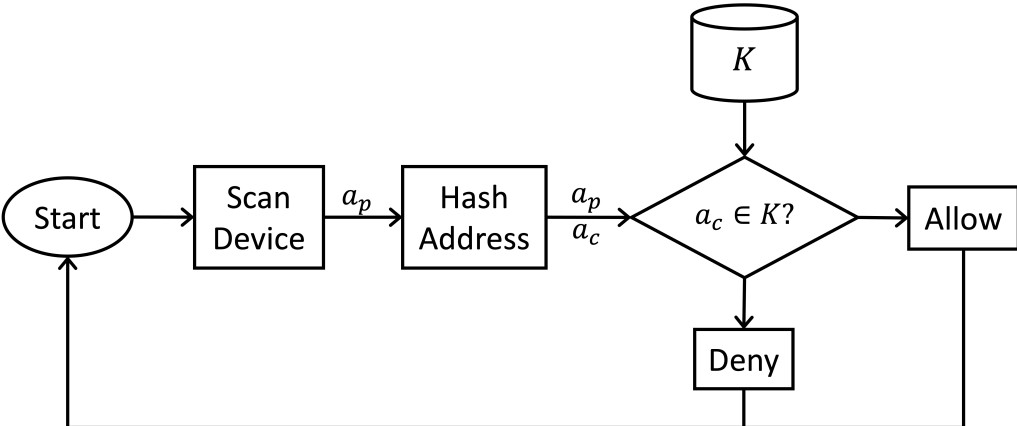

**Figure 12.** A flowchart illustrating how the proposed embedded countermeasure against MQTT-based MitM and masquerade attacks works.

The implemented countermeasure adds a second security layer to prevent the executed DoS attack, and prevents MitM and masquerade attacks by storing hashed and plain-text IP and MAC addresses in a database and feeding such information of a source to a packet-filtering firewall along with the command. This way, malicious data will be blocked from reaching the ZED, as opposed to relying on the original system, which ultimately denied accessibility to authorized users. Due to the power and memory limitations of IoT systems, energy consumption, processing time, and performance overhead of an IDS are vital performance metrics to consider when designing an IDS for an IoT-based environment.

Given the challenges in end-user IP spoofing detection, the following recommendations are proposed to reduce spoofing and mitigate DoS, MitM, and masquerade attacks:

1. *Packet filtering*: examining IP packets for each device or user joining a network, specifically the header of each IP packet to authenticate the source. If the IP address does not match the source, the packet will be restricted from completing the connection. That is, packets sourced from outside of the local network but claim to be within are to be rejected.
2. *Access Control Lists (ACLs)*: denying private IP addresses on downstream interface.
3. *Enabling encryption sessions*: trusted hosts outside the IoT network can securely communicate with local hosts when enabling router encryption sessions.

A general approach to secure communication is the placement of crucial and critical computing resources behind a firewall, and the installation of virus and malware protection to run at boot. At the user end, an effective security mitigation method is two-factor authentication. Incorporating another vector of authentication greatly decreases the chances of unauthorized access to networks. Lastly, it is important to run a security scan as soon as an encrypted connection is expected but nonexistent, as it may be a clear sign of an attack. Blockchain is being considered the next frontier in countermeasures for securing IoT devices and networks against integrity attacks [51–53]. In response to the resource constraints in IoT networks, compression-aware authorization is proposed [54].

## 7. Reflection on the Experiments

As industries continue with digital transformations, IoT solutions must account for increased productivity and cost-efficiency demand, as such, the digital revolution has increased the dependence on technology. In this light, the impact of exposed ZigBee keys was profound: offering malicious actors unauthorized accessibility to sensitive data and an opportunity to expose, destroy or manipulate it. Spreading the attack to CoAP, it is evident that closing the gaps in IoT security remains a challenge due to power, bandwidth, processing, and cost limitations in connected devices.

In both experiments, it must be remarked that the remote had not been directly compromised by any of the attacks. Since it does not require a gateway connection after initial configuration, it was not targeted in the DoS attack. Instead, its messages were modified after reaching the ZED, yet the firmware and device as a whole continued their secure operation. While it is not usually an issue, complete reliance on IoT, an emerging technology with many vulnerabilities, can magnify the impact of successful data breaches. Aside from that, the lack of industry foresight has exposed organizations and manufacturers to cyberthreats, especially with overlooking investments in IoT device security.

ZigBee network security could be improved through patching vulnerabilities in ZCs such that intrusion detection and prevention processes are carried out immediately and efficiently. Although the AES algorithm provides encryption of sufficient robustness, its security is dependent upon the secrecy of the encryption keys, which can be breached during initialization or distribution. Thus, upgraded ZigBee security should not be solely based on key secrecy. Namely, the firmware of ZCs is to be upgraded such that a firewall filters traffic in order to prevent flooding and data from suspicious sources from reaching ZEDs. In the first countermeasure, the IDS was programmed to detect unusual flooding of data aiming to keep the device off, which was only achievable in case of an attack due to the existence of a single switching button. Comparably, other test cases could be implemented by different solution providers to be specifically tailored to their products. If the first IDS, which suggests the usage of an MQTT broker, is infeasible, one can install the MQTT broker software on the existing network coordinator device (ZC previously). This way, the MQTT broker can use ZigBee2MQTT to communicate with ZEDs. Alternatively, the second IDS remains a good substitute for outsider DoS threats. As demonstrated by the algorithm, a joining device will be authenticated first before gaining authority over any ZED in the network. Extending this to DoS, an attacker will be unable to join if their identity does not match any legitimate one retrieved from the database. Therefore, they will be unable to

flood the network or cause it to crash. Yet, this does not necessarily prevent DoS attacks from insider sources. In other words, the authentication algorithm will be irrelevant if the attack originates from within the targeted network. In this case, MQTT broker software will be necessary to provide better protection for the system.

Considering the above mitigation techniques, it remains a future goal to fully implement, test, and demonstrate the performance achieved by the proposed techniques, as the main focus of this work is not securing the ZigBee protocol, but rather exploiting its vulnerabilities and showcasing the consequences, which generalize to other ZigBee-enabled products. However, it can be argued that the mitigation techniques are not expensive to implement. In the first IDS, ZigBee2MQTT is widely available as a low-cost USB dongle, which is advantageous since it does not require permanent storage on an MQTT device. Therefore, it does not require storage space compensation. In the second IDS, the database will keep track of information related to a small, finite number of devices in the second mitigation technique. This is supported by the practical limitation of ZigBee-connected devices allowed by a ZC. For instance, the IKEA TRÅDFRI gateway allows up to 10 user devices to connect to it through the mobile application. As such, retrieving and authenticating users should require a small storage space, which is suitable for storage-constrained IoT devices. Finally, the lightweight implementations should not cause conspicuous processing delays, although this can only be proven when the IDSs are fully developed and implemented.

The security of IoT-based smart home consumer devices is an important problem [55–59]. Although our experiments in this work have focused on IKEA products to demonstrate vulnerabilities, the security of other IoT-based smart home devices has been ranked in other works [60]. Our work has also not tackled the privacy problem of IoT devices. Privacy preservation has been proposed as one of the promising approaches to addressing the privacy of IoT devices [61].

## 8. Conclusions and Future Work

The paper has explored the vulnerabilities in ZigBee and CoAP protocols in a commercial product. It has been demonstrated that these vulnerabilities can be exploited with well-known attack scenarios. Mitigation mechanisms against these attacks have also been suggested and shown and, if successfully implemented, the proposed attacks can be minimized. The implications of the findings of this work are not only limited to smart lamps. Any product that uses ZigBee and CoAP can be exploited in a similar way, as demonstrated by attacks in the experiments.

One of the proposed countermeasures is a generalized IDS that mitigates Remote AT Commands targeting ZEDs by having the ZC keep track of IP and MAC addresses of all authorized devices that are present in the network, and only allowing messages of authorized sources to be passed to ZEDs. Correspondingly, securing other nodes in the IoT network is mandatory. Appropriate CoAP user-end security measures can include password-based authentication, multi-factor authentication, or biometric authentication. It is to be noted that none of the aforementioned authentication methods were implemented in the IKEA Home Smart application, which was used in the experiments. The absence of these countermeasures grants malicious actors the opportunity of joining the IoT network by compromising CoAP user-end nodes instead of the ZC. Thus, in future work, the proposed countermeasures will be evaluated for their effectiveness.

**Author Contributions:** Conceptualization, N.H. and A.N.; methodology, N.H. and A.N.; software, N.H.; validation, N.H. and A.N.; formal analysis, N.H.; investigation, N.H.; resources, A.N.; data curation, N.H.; writing—original draft preparation, N.H.; writing—review and editing, N.H. and A.N.; visualization, N.H. and A.N.; supervision, A.N.; project administration, A.N.; funding acquisition, A.N. All authors have read and agreed to the published version of the manuscript.

**Funding:** This research received no external funding.

**Institutional Review Board Statement:** Not applicable.

**Informed Consent Statement:** Not applicable.

**Data Availability Statement:** Not applicable.

**Conflicts of Interest:** The authors declare no conflict of interest.

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
