# Peer review of "Living in the Dark: MQTT-Based Exploitation of IoT Security Vulnerabilities in ZigBee Networks for Smart Lighting Control"

_2624-831X, doi:10.3390/iot3040024_

Round 1

Reviewer 1 Report

1. Is there a partnership with IKEA for this research? Is there an agreement?

2. Why did the author wish to deal with ZigBee-based applications?

3. The motivation and purpose of the research should be more clearly stated.

4. The ZigBee communication protocol itself has the key and related encryption technology. What is the contribution of this research?

5. Lack of some actual equipment photos or experimental photos.

6. The authors should provide some actual test data to demonstrate the performance achieved by the proposed algorithm.

Author Response

Please note that each comment has been labelled as RxCy, where x is the reviewer number and y is the comment number in the received feedback.

Reviewer 2 Report

Thank you for you hard work and very clear presentation. Your paper is very readable (although some proofreading would help) and clear. Your methods are well presented and should be reproducible by a knowledgeable reader. However, you hacked one system and I was left wondering if you'd discovered anything truly new about the Zigbee protocol.

I don't think anyone would be surprised that you found vulnerabilities and exploited an IoT device. What exactly is new about your approach or discoveries? Do your methods generalize to many Zigbee systems? How expensive are your remediations? Would they apply generally? Have these vulnerabilities been responsibly disclosed to the manufacturer? And I have a few questions and nits:

Message Queue Telemetry Transport (MQTT) should be defined before first use.

Lines 137-138: You say, "a pre-configured key is used to encrypt the network key after the distribution. Despite this, there remains a huge risk of having a single shared encryption key on all the network". Is this risk a likely feature due to a small key space, or a hypothetical and very unlikely occurrence?

Line 321, "This paper utilizes identified ZigBee and CoAP security vulnerabilities to exploit an IoT network" So how is this work truly novel? You are exploiting known vulnerabilities, not discovering new ones. 

Line 445: "the Trust Center link key, which is almost universal for all ZigBee networks" Is it truly universal, drawn from a small set of known keys, or just easy to predict? Can users change it? If so, this would be a good recommendation.

Lines 449-451: "If the Network Key has been changed, the attacker can retrieve the Transport Key from Wireshark when a ZigBee device joins the network. From the ZigBee Network Layer Data sub-tree, the Key will be unencrypted and exposed to the attacker." Passing keys in the clear is a major bug. Is this common to all Zigbee implementations? If so, your attack generalizes. But I think you mean that this key is encrypted by an easy to guess or constant-valued key, not passed exactly in the clear.

Once you've gotten to this point, it is game over. Confidentiality and Integrity are lost. The only thing that remains is to interrupt Availability, which is what the rest of the attack part is about.

Line 523: "unauthorized access to the IoT system will be possible through the acquisition of the ZigBee installation code." Is this what is meant earlier by the code being "almost universal?" Do you mean that having this code you can easily derive the key? How hard is that? How easy is it to steal this code?

One thing I would like to know is how difficult this attack would be to implement in the real world. Can this attack be carried out without physical access over the internet? What are the real-world consequences that can translate into dollars or safety that you could articulate?

I like how you finished the paper by very specifically proposing fixes to the system you broke. This is a real contribution to the manufacturer, and, if generalized, could be a contribution to the state of the art as well. Having to modify the firmware to effect some of these changes puts those parts out of the reach of most system owners though.

Please consider how you could make more of a statement about the state of the art, not about one manufacturer's product. Your lessons learned are good, but it is unclear whether they are needed everywhere or just for this system. I think your paper's contribution would be greatly enhanced by showing that the hack and mitigations are general and widely applicable.

Author Response

(The authors gave the same response as above.)

Reviewer 3 Report

The paper has explored the vulnerabilities in ZigBee and CoAP protocols in a commercial product. The paper was good defined and demonstrated excellent overview of the three security features. The author presents the security vulnerability that exploit the IoT network, good description of the attack on smart lighting and including DoS, MitM and Masquerade. Some of the figures are confusing and vague such as Fig 10 and 11, need further clarification. 

Author Response

(The authors gave the same response as above.)

Reviewer 4 Report

In the introduction it is unclear what constrained devices you have in
mind. While the contribution addresses the IKEA Trådfri, the
introduction talks about IoT devices' large attack surfaces and
potential need of integrating "sophisticated firewalls or anti-malware
software". Considering that the smart bulbs MCU implements a very
limited set of services and therefore has only a few ports open, the
attack surface seems very limited compared to multi-purpose devices
such as, e.g., a Raspberry Pi with a full-fledged multi-user operating
system.

# Detailed Comments

* Title: A great part of this paper talks specifically about security
  of MQTT and Zigbee. Affecting CoAP is a side-effect and therefore
  the title should address MQTT rather than CoAP.

* p1, l24f: Identifiers of IoT nodes do not have to be unique at
  all. There are many protocols and implementations that handle it
  this way (to a certain degree of uniqueness) but there is no
  technical requirement of unique identifiers by (sensor) networks at
  all. Usually, these "unique identifiers" are brought in by lack of
  imagination, i.e., poor system design.

  I suggest removing that part from the sentence: "Nodes in an IoT
  network provide automatic…"

* p1, l29f: This sentence does not parse: "[…] IoT devices and
  networks are of a large attack surface despite the valuable
  accessibility,[…]".

* p2, l40f: The sentence in l41ff ("However…") contradicts the
  statement in line 40f.

* p2, l44ff: The protocol names are missing the article (→ "The
  Constrained […] the User Datagram […] the Hypertext […]"

* p2, l50: Why are AT commands mentioned together with the description
  of CoAP? This is not only unrelated but lacks any motivation of why
  AT commands are considered not secure by the authors. (→ missing
  data transparency of data frames)

* p2, l60: "gateway IP" → "gateway IP address"

* p3, l93: ZigBee is not a wireless standard but a brand for a
  protocol silo. The PHY and MAC layers of this stack are based on
  IEEE 802.15.4.

* p3, 99: "ZEDs (ZED)" → "ZigBee End Devices (ZEDs)"

* p4, Section 2.2: The classic reference for CoAP is missing:
  C. Bormann, A. P. Castellani and Z. Shelby, "CoAP: An Application
  Protocol for Billions of Tiny Internet Nodes," in IEEE Internet
  Computing, vol. 16, no. 2, pp. 62-67, March-April 2012, doi:
  10.1109/MIC.2012.29.

* p4, l173: BLE is not part of the CoAP specification, so this
  statement here is misleading.

* (p4, l176f: HTTP mapping has been stressed during the initial
  standaridzation phase of CoAP but the important additional features
  of CoAP cannot be mapped easily to HTTP. Therefore, stateless HTTP
  is usually not emphasized.)

* p5, Section 2.2.2: The possible security mechanisms lack OSCORE (RFC
  8613) on the application layer.

* p5, l186: DTLS by itself does not provide non-repudiation.

* p5, l186: The articles are missing for the protocols.

* p7, Section 3.2: There is also draft-ietf-lwig-security-protocol-comparison

Author Response

(The authors gave the same response as above.)

Round 2

Reviewer 1 Report

My comments have been addressed.